# Adaptation to genome decay in the structure of the smallest eukaryotic ribosome

David Nicholson [1], Marco Salamina[2], Johan Panek [2], Karla Helena-Bueno[2], Charlotte R. Brown [2], Robert P. Hirt [2], Neil A. Ranson [1✉] & Sergey V. Melnikov [2,3✉]

The evolution of microbial parasites involves the counterplay between natural selection forcing parasites to improve and genetic drifts forcing parasites to lose genes and accumulate deleterious mutations. Here, to understand how this counterplay occurs at the scale of individual macromolecules, we describe cryo-EM structure of ribosomes from *Encephalitozoon cuniculi*, a eukaryote with one of the smallest genomes in nature. The extreme rRNA reduction in *E. cuniculi* ribosomes is accompanied with unparalleled structural changes, such as the evolution of previously unknown molten rRNA linkers and bulgeless rRNA. Furthermore, *E. cuniculi* ribosomes withstand the loss of rRNA and protein segments by evolving an ability to use small molecules as structural mimics of degenerated rRNA and protein segments. Overall, we show that the molecular structures long viewed as reduced, degenerated, and suffering from debilitating mutations possess an array of compensatory mechanisms that allow them to remain active despite the extreme molecular reduction.

[1] Astbury Centre for Structural Molecular Biology, School of Molecular & Cellular Biology, Faculty of Biological Sciences, University of Leeds, Leeds LS2 9JT, UK. [2] Biosciences Institute, Newcastle University, Newcastle upon Tyne NE2 4HH, UK. [3] Faculty of Medical Sciences, Newcastle University, Newcastle upon Tyne NE2 4HH, UK. ✉email: n.a.ranson@leeds.ac.uk; sergey.melnikov@ncl.ac.uk

Because most branches of microbial parasites possess unique molecular tools to exploit their hosts, we frequently need to develop distinct therapies against distinct groups of parasites[1,2]. However, emerging evidence shows that some aspects of parasite evolution are convergent and largely predictable, suggesting a potential basis for broad-range therapies against microbial parasites[3–9].

Previous work determined one common evolutionary tendency in microbial parasites known as genome reduction or genome decay[10–13]. Current studies indicate that—when a microorganism abandons its free-living lifestyle to become an intracellular parasite (or an endosymbiont)—its genome undergoes a slow but startling metamorphosis over a period of millions of years[9,11]. In a process known as genome decay, microbial parasites accumulate deleterious mutations that transform many formerly essential genes into pseudogenes, leading to gradual gene loss and mutational meltdown[14,15]. This meltdown can eliminate up to ~95% of genes in the most ancient intracellular organisms compared to closely related free-living species[16–18]. Thus, the evolution of intracellular parasites involves a tug of war of two opposing forces—with Darwinian natural selection pushing parasites to improve, and genome decay pushing parasites to fade into oblivion. How successful parasites emerge from this tug of war, preserving the activity of their molecular structures, remains unclear.

Although the mechanism of genome decay is not entirely clear, it appears to primarily stem from frequent genetic drifts. Because parasites live in small, asexual and genetically bottle-necked populations, they cannot effectively eliminate deleterious mutations that sporadically occur during DNA replication. This causes irreversible accumulation of deleterious mutations and reduction of parasite genomes[18]. Thus, it is not that parasites lose only those genes that are no longer essential for their survival in the intracellular context. It is that parasites populations cannot effectively eliminate sporadic deleterious mutations, causing accumulation of these mutations throughout their genomes, including their most essential genes[10–13].

Most of what we currently know about genome reduction is solely based on comparisons of genome sequences, paying lesser attention to changes in the actual molecules that perform housekeeping functions and serve as potential targets for drugs. Comparative studies revealed that the burden of deleterious mutations in intracellular microorganisms appears to render proteins and nucleic acids prone to misfolding and aggregation, making them more dependent on chaperons and hypersensitive to heat[19–23]. Furthermore, distinct parasites—sometimes separated by as much as 2.5 billion years of independent evolution—have experienced a similar loss of quality-control centers in their protein synthesis[5,6] and DNA repair machineries[24]. However, we know little about the impact of intracellular lifestyles on all other properties of cellular macromolecules, including molecular adaptations to the ever-increasing burden of deleterious mutations.

In this work, seeking to better understand the evolution of proteins and nucleic acids in intracellular microorganisms, we determined the structure of ribosomes from the intracellular parasite *Encephalitozoon cuniculi*. *E. cuniculi* is a fungi-like organism, which belongs to the group of parasites microsporidia, which possess anomalously small eukaryotic genomes and are therefore used as model organisms to study genome decay[25–30]. Recently, cryo-EM structures of ribosomes were determined for microsporidians with moderately reduced genomes, *Paranosema locustae* and *Vairimorpha necatrix*[31,32] (~3.2 Mb genomes). These structures showed that the loss of some rRNA expansions is compensated by evolving new contacts between adjacent ribosomal proteins or by the acquisition of a new ribosomal protein, msL1[31,32]. *Encephalitozoon* species (~2.5 Mb genomes), along with their closest relatives *Ordospora*, show the ultimate degree of genome reduction in

eukaryotes—they have less than 2,000 protein-coding genes and their ribosomes were predicted to lack not only rRNA expansion segments (rRNA segments that distinguish eukaryotic ribosomes from bacterial ribosomes) but also four ribosomal proteins due to the absence of their homologs in the *E. cuniculi* genome[26–28]. Therefore, we reasoned that *E. cuniculi* ribosomes could reveal previously unknown strategies of molecular adaptation to genome decay.

Our cryo-EM structure represents the smallest eukaryotic cytoplasmic ribosome to be characterized and provides insights into how the ultimate degree of genome reduction affects the structure, assembly and evolution of indispensable molecular machineries of a cell. We found that *E. cuniculi* ribosomes defy many widely conserved principles of RNA folding and ribosome assembly, and we identified a new, previously unknown, ribosomal protein. Most unexpectedly, we show that microsporidian ribosomes have evolved the ability to bind small molecules and hypothesize that truncations in rRNA and proteins trigger evolutionary innovations that can ultimately endow ribosomes with beneficial qualities.

## Results

**Isolation of *E. cuniculi* ribosomes.** To improve our understanding of the evolution of proteins and nucleic acids in intracellular organisms, we set out to isolate *E. cuniculi* spores from infected mammalian cell cultures to purify their ribosomes and determine the structure of these ribosomes. Large quantities of microsporidian parasites are challenging to produce because microsporidians cannot be cultured in a growth medium. Instead, they grow and reproduce only inside their host cells. Therefore, to produce *E. cuniculi* biomass for the ribosome purification, we infected mammalian kidney cell line RK13 with *E. cuniculi* spores and cultivated these infected cells for several weeks to allow for *E. cuniculi* to grow and reproduce. Using approximately half a square meter of the infected cells' monolayer, we could purify about 300 mg of microsporidian spores and use them for ribosome isolation. We then broke the purified spores with glass beads and isolated crude ribosomes using a stepwise fractionation of lysates with polyethylene glycol. This allowed us to obtain approximately 300 μg of crude *E. cuniculi* ribosomes for structural analyses.

**The structure of the smallest eukaryotic cytoplasmic ribosome.** We then used the obtained ribosome sample to collect cryo-EM images and process these images with masks corresponding to the large ribosomal subunit, the head of the small subunit, and the body of the small subunit. In doing so, we collected snapshots of ~108,000 ribosomal particles and calculated cryo-EM maps at 2.7 Å resolution (Supplementary Figs. 1–3). We then used the cryo-EM maps to build the model of rRNA, ribosomal proteins and the hibernation factor Mdf1 bound to *E. cuniculi* ribosomes (Fig. 1a, b).

Compared to the previously determined structures of *V. necatrix* and *P. locustae* ribosomes (both structures represent the same family of *Nosematidae* microsporidians and are very similar to each other)[31,32], *E. cuniculi* ribosomes underwent further degeneration of numerous rRNA and proteins segments (Supplementary Figs. 4–6). In rRNA, the most prominent changes include the complete loss of the 25 S rRNA expansion segment ES12$^L$ and partial degeneration of helices h39, h41, and H18 (Fig. 1c, Supplementary Fig. 4). In ribosomal proteins, the most prominent changes include the complete loss of protein eS30 and truncations in proteins eL8, eL13, eL18, eL22, eL29, eL40, uS3, uS9, uS14, uS17, and eS7 (Supplementary Figs. 4, 5).

Thus, the extreme genome reduction in *Encephalotozoon/Ordospora* species is reflected in the structure of their ribosomes:

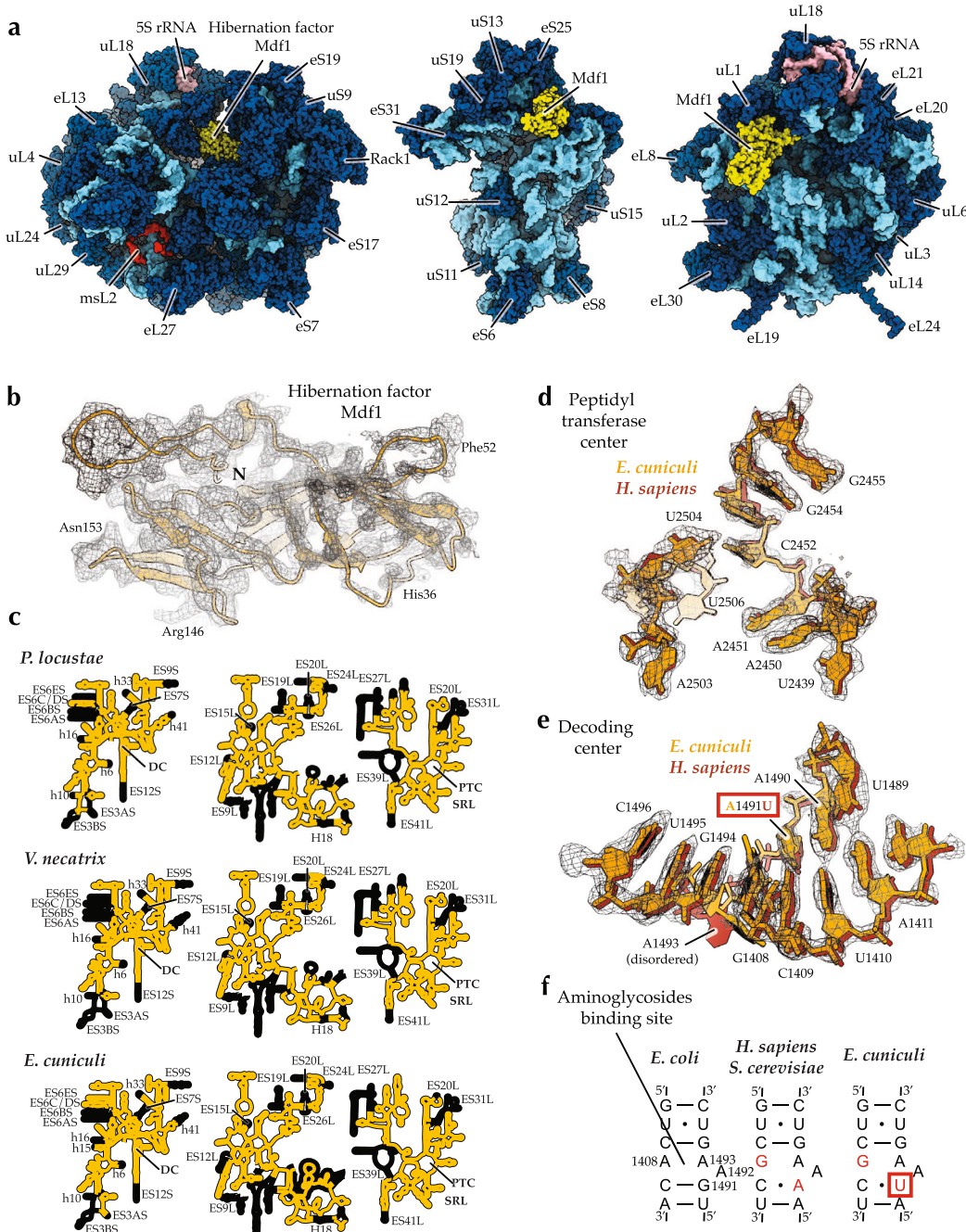

**Fig. 1 Electron microscopy reveals the structure of the miniaturized ribosomes from the human pathogen Encephalitozoon cuniculi. a** The structure of *E. cuniculi* ribosomes in complex with the hibernation factor Mdf1 (pdb id 7QEP). **b** The map of the hibernation factor Mdf1 bound to *E. cuniculi* ribosomes. **c** Secondary structure diagrams compare rRNA reduction in microsporidian species with known ribosome structures. The panels indicate the location of rRNA expansion segments (ES) and ribosomal active centers, including the decoding site (DC), the sarcin-ricin loop (SRL), and the peptidyl-transferase center (PTC). **d** The electron density corresponding to the peptidyl-transferase center of E. cuniculi ribosomes shows that this catalytic site has the same structure in the parasite *E. cuniculi* and its hosts, including *H. sapiens*. **e**, **f** The electron density corresponding to the decoding center (**e**) and schematic structures of the decoding center (**f**) illustrate that *E. cuniculi* have U1491 residue instead of A1491 (*E. coli* numbering) in many other eukaryotes. This variation suggests that *E. cuniculi* may have sensitivity to the antibiotics targeting this active site.

*E. cuniculi* ribosomes have experienced the most drastic loss of protein content among eukaryotic cytoplasmic ribosomes to be structurally characterized, and they are devoid of even those rRNA and protein segments that are widely conserved not only in eukaryotes but across the three domains of life. The structure of *E. cuniculi* ribosomes provided the first molecular model of these changes and revealed evolutionary events that were overlooked by both comparative genomics and structural studies of molecules from intracellular organisms (Supplementary Fig. 7). Below, we describe each of these events, along with their possible evolutionary origin and their potential impact on ribosome function.

**Variations in the decoding center and the drug-binding site.** We next observed that, aside from large rRNA truncations,

*E. cuniculi* ribosomes possess rRNA variations in one of their active sites. While the peptidyl-transferase center of *E. cuniculi* ribosomes has the same structure as in other eukaryotic ribosomes (Fig. 1d), the decoding center differs due to the sequence variation in the nucleotide 1491 (*E. coli* numbering, Fig.1e, f). This observation is important because the decoding site of eukaryotic ribosomes typically contains residues G408 and A1491, compared to bacteria-type residues A1408 and G1491. And this variation underlies different sensitivity of bacterial and eukaryotic ribosomes to the aminoglycoside family of ribosome-targeting antibiotics and other small molecules targeting the decoding site[33–35]. In the decoding site of *E. cuniculi* ribosomes, the A1491 residue is replaced with U1491, potentially creating a unique binding interface for small molecules targeting this active center. The same A14901 variation is present in other microsporidians, such as *P. locustae* and *V. necatrix*, suggesting its wide occurrence in microsporidian species (Fig. 1f).

**Ribosome hibernation by the factor Mdf1.** Because our samples of *E. cuniculi* ribosomes were isolated from metabolically inactive spores, we tested the cryo-EM maps of *E. cuniculi* for the presence of previously described hibernation factors that bind ribosomes under stress or starvation conditions[31,32,36–38]. We docked previously determined structures of hibernating ribosomes in the cryo-EM maps of *E. cuniculi* ribosomes. For this docking, we used *Saccharomyces cerevisiae* ribosomes in complex with the hibernation factor Stm1[38], *P. locustae* ribosomes in complex with the factor Lso2[32], and *V. necatrix* ribosomes in complex with factors Mdf1 and Mdf2[31]. In so doing, we found the cryo-EM density corresponding to the hibernation factor Mdf1. Similar to Mdf1 binding to *V. necatrix* ribosomes, Mdf1 also binds *E. cuniculi* ribosomes, where it blocks the ribosomal E site, possibly helping inactivate ribosomes when parasites sporulate and become metabolically inactive (Fig. 2).

Our structure, however, revealed a previously unknown contact between Mdf1 and the ribosomal L1-stalk (the part of the ribosome that helps release deacylated tRNAs from the ribosome during protein synthesis). Specifically, Mdf1 exploits the same contacts as the elbow-segment of deacylated tRNA molecules (Fig. 2). This previously unknown molecular mimicry suggests that Mdf1 dissociates from the ribosome using the same mechanism as deacetylated tRNAs, explaining how ribosomes can remove this hibernation factor to reactivate protein synthesis.

**E. cuniculi rRNA trades optimal folding for minimal length.** While building the rRNA model, we found that *E. cuniculi* ribosomes possess anomalously folded rRNA segments, which we termed molten rRNA (Fig. 3). In ribosomes across the three domains of life, rRNA folds into structures in which most rRNA bases are either base-paired and stacked with each other or interact with ribosomal proteins[38–40]. However, in *E. cuniculi* ribosomes, the rRNA appears to defy this folding principle by transforming some of their helices into unfolded rRNA stretches.

The most striking example of this evolutionary transformation can be observed in the helix H18 of the 25 S rRNA (Fig. 3). In species ranging from *E. coli* to humans, the base of this rRNA helix contains 24-32 nucleotides that form a slightly irregular helical structure. In the previously determined structures of ribosomes from *V. necatrix* and *P. locustae*[31,32] the base of helix H18 is partially unwound yet the base-pairing of nucleotides is preserved. In *E. cuniculi*, however, this rRNA segment is turned into the minimal-length linkers $^{228}$UUUGU$^{232}$ and $^{301}$UUUUUUU$^{307}$. Unlike typical rRNA segments, these uridine-rich linkers are neither folded into a helix nor they are involved in extensive contacts with ribosomal proteins. Instead,

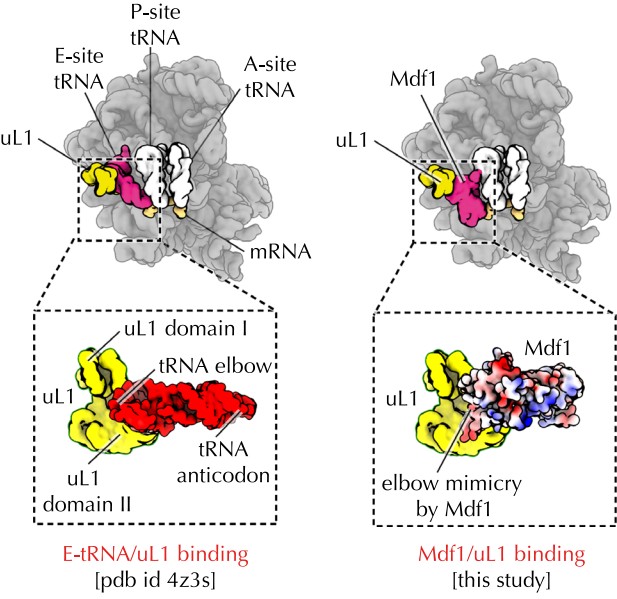

Fig. 2 Ribosome hibernation factor Mdf1 mimics deacetylated tRNAs. Mdf1 blocks the ribosomal E site, which appears to help inactivate ribosomes when parasites sporulate and become metabolically inactive. In the structure of *E. cuniculi* ribosomes, we found that Mdf1 forms a previously unknown contact with the ribosomal L1-stalk (the part of the ribosome that helps release deacylated tRNAs from the ribosome during protein synthesis). These contacts suggest that Mdf1 dissociates from the ribosome using the same mechanism as deacetylated tRNAs, providing a possible explanation of how ribosomes can remove Mdf1 to reactivate protein synthesis.

they adopt a solvent-exposed and fully unfolded structure in which rRNA strands are stretched into an almost straight line. This stretched conformation explains how *E. cuniculi* can use just 12 RNA bases to fill the 33 Å-long gap between rRNA helices H16 and H18—while other species require at least twice as many rRNA bases to fill this gap.

Thus, we could show that at the expense of energetically unfavorable folding, microsporidian parasites have invented a strategy to reduce even those rRNA segments that remain widely conserved across species from the three domains of life. Apparently, by accumulating mutations that transform rRNA helices into short poly-U linkers, *E. cuniculi* could evolve unusual rRNA segments that comprise the minimum possible number of nucleotides that is required to connect distant segments of rRNA. This helps explain how microsporidia have accomplished the phenomenal reduction of their essential molecular structure without losing its structural and functional integrity.

**E. cuniculi rRNA is stripped of highly conserved bulges.** Another anomalous feature of *E. cuniculi* rRNA is the emerging of bulgeless rRNA (Fig. 4). Bulges are non-base-paired nucleotides that flip out from RNA helices rather than being buried inside a helix[41]. Most rRNA bulges serve as a molecular glue by helping to bind adjacent ribosomal proteins or other rRNA segments. Some bulges serve as a hinge that allows rRNA helices to bend and adopt an optimal folding for productive protein synthesis[41].

Strikingly, we observed that *E. cuniculi* ribosomes lack most of the rRNA bulges found in other species, including more than 30 bulges that are conserved in other eukaryotes (Fig. 4a). This loss eliminates many contacts between ribosomal subunits and adjacent rRNA helices, occasionally creating large hollow voids

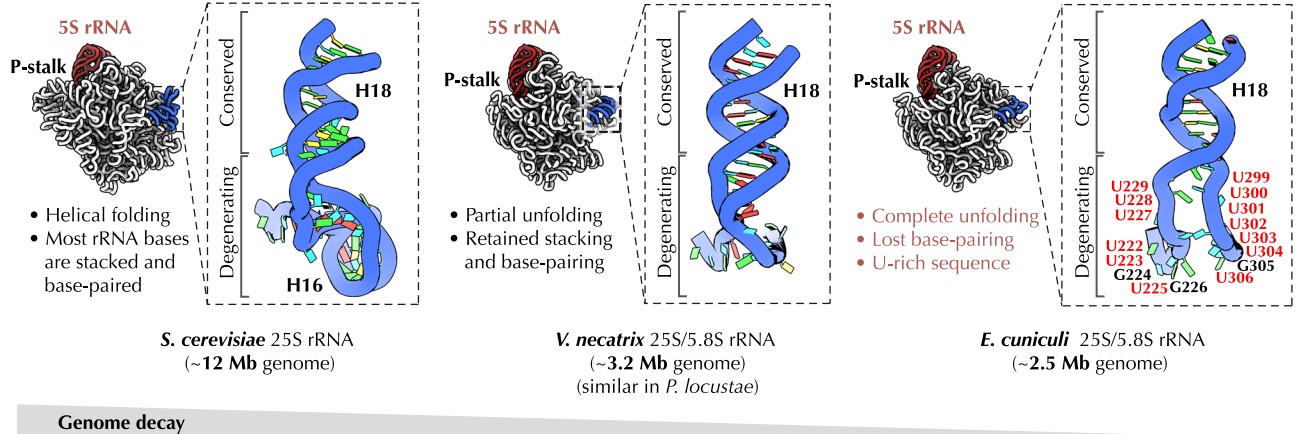

**Fig. 3 Microsporidian rRNA trades favorable folding for minimal length.** Structure of the helix H18 of the 25 S rRNA in *S. cerevisiae*, *V. necatrix*, and *E. cuniculi*. Typically, in ribosomes across the three domains of life, this linker is folded into an RNA helix, which comprises between 24 and 34 residues. By contrast, in microsporidia this rRNA linker is being progressively reduced to two, single-stranded uridine-rich linkers that comprise just 12 residues. Most of these residues are exposed to the solvent. This figure illustrates that microsporidian parasites appear to defy a common principle of rRNA folding, in which rRNA bases are typically paired with other bases or involved in rRNA–protein interactions. In microsporidia, some rRNA segments adopt unfavorable folding, in which former rRNA helices are turned into single-stranded segments that are stretched out almost into a straight line. Having these unusual stretches allows microsporidian rRNA to connect distant segments of rRNA using the minimal number of RNA bases.

within the ribosome interior, making *E. cuniculi* ribosomes more porous compared to the more conventional ribosomes (Fig. 4b). Notably, we found that most of these bulges are also lost in the previously determined structures of *V. necatrix* and *P. locustae* ribosomes, which was overlooked by previous structural analyses[31,32].

Occasionally, the loss of rRNA bulges is accompanied by the evolution of new bulges near the lost ones. For example, the ribosomal P-stalk contains a bulge U1208 (in *S. cerevisiae*) that is conserved from *E. coli* to humans and is therefore estimated to be 3.5 billion years old. During protein synthesis, this bulge helps the P-stalk to move between open and closed conformations so that the ribosome can recruit translation factors and deliver them to the active site[42]. In *E. cuniculi* ribosomes this bulge is missing; however, a new bulge (G883) is located just three base pairs away, possibly helping restore the optimal flexibility of the P-stalk (Fig. 4c).

Our finding of bulgeless rRNA shows that rRNA minimization is not limited to the loss of rRNA elements on the surface of the ribosome but may affect the very core of the ribosome, creating a parasite-specific molecular defect that has not been observed in free-living species.

**Small molecules as ribosomal building blocks**. Having modelled canonical ribosomal proteins and rRNA, we found three segments of the cryo-EM map not accounted for by the conventional ribosome components. Two of these segments had a size of small molecules (Fig. 5, Supplementary Fig. 8). The first segment was sandwiched between ribosomal proteins uL15 and eL18 at a location normally occupied by the eL18 C-terminal truncated in *E. cuniculi*. Although we could not determine the identity of this molecule, the size and shape of this density island would be explained well by the presence of a spermidine molecule. Its binding to the ribosome is stabilized by microsporidia-specific mutations in protein uL15 (Asp51 and Arg56), which appear to increase the ribosome affinity to this small molecule as they allow uL15 to wrap around this small molecule in the ribosome structure (Supplementary Fig. 8, Supplementary Data 1, 2).

The second small molecule density was located at the interface of ribosomal proteins uL9 and eL30 (Fig. 5a). This interface was previously described in the structure of *S. cerevisiae* ribosomes as

a binding site of the 25 S rRNA nucleotide A3186 (part of the rRNA expansion segment ES39L)[38]. In *P. locustae* ribosomes, where ES39L is degenerated, this interface was shown to bind an unidentified single nucleotide[31], and it was hypothesized that this nucleotide represents the ultimate form of rRNA reduction in which the ~130-230 base-long rRNA expansion ES39L was reduced to a single nucleotide[32,43]. Our cryo-EM maps confirmed the idea that the density can be accounted for by a nucleotide. However, the higher resolution of our structure revealed that this nucleotide is an extra-ribosomal molecule, likely AMP (Fig. 5a, b).

We next asked whether the nucleotide-binding site has been evolved or preexisted in *E. cuniculi* ribosomes. Because the nucleotide-binding is primarily mediated by residues Phe170 and Lys172 in the ribosomal protein eL30, we assessed the conservation of these residues in 4,396 representative eukaryotes. Similar to the aforementioned case of uL15, we found that Phe170 and Lys172 residues are highly conserved only in canonical microsporidia, but are absent in other eukaryotes, including non-canonical microsporidia *Mitosporidium* and *Amphiamblys* in which the rRNA ES39L segment is not reduced[44–46] (Fig. 5c–e).

Collectively, these data supported the idea that *E. cuniculi*, and possibly other canonical microsporidians, have evolved the ability to effectively trap abundant small metabolites in their ribosome structures in order to compensate for the rRNA and protein reduction. In doing so, they have evolved the unique ability to bind extra-ribosomal nucleotides, illustrating a previously unknown and ingenious ability of parasitic molecular structures to compensate their degeneration by trapping small abundant metabolites and using them as structural mimics of degenerated RNA and protein segments.

**Diversified protein content in microsporidian ribosomes**. The third unmodeled segment of the cryo-EM map we found within the large ribosomal subunit. The relatively high resolution of our maps (2.6 Å) revealed that this density belongs to a protein with a unique combination of bulky side chains residues, which allowed us to identify this density as a previously unknown ribosomal protein, which we termed msL2 (microsporidia-specific protein L2) (Methods, Fig. 6). Our homology search revealed that msL2 is conserved in the microsporidian branch of *Encephalitozoon* and

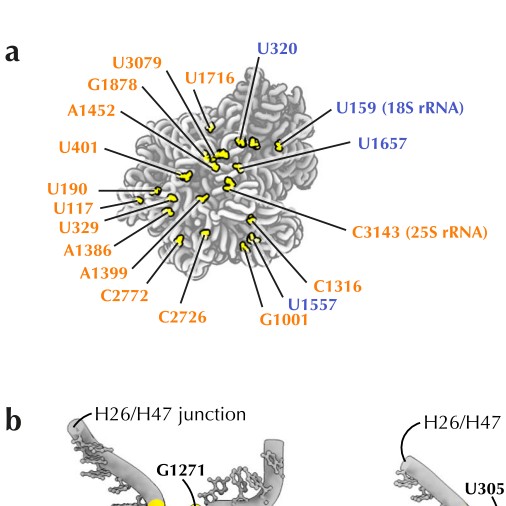

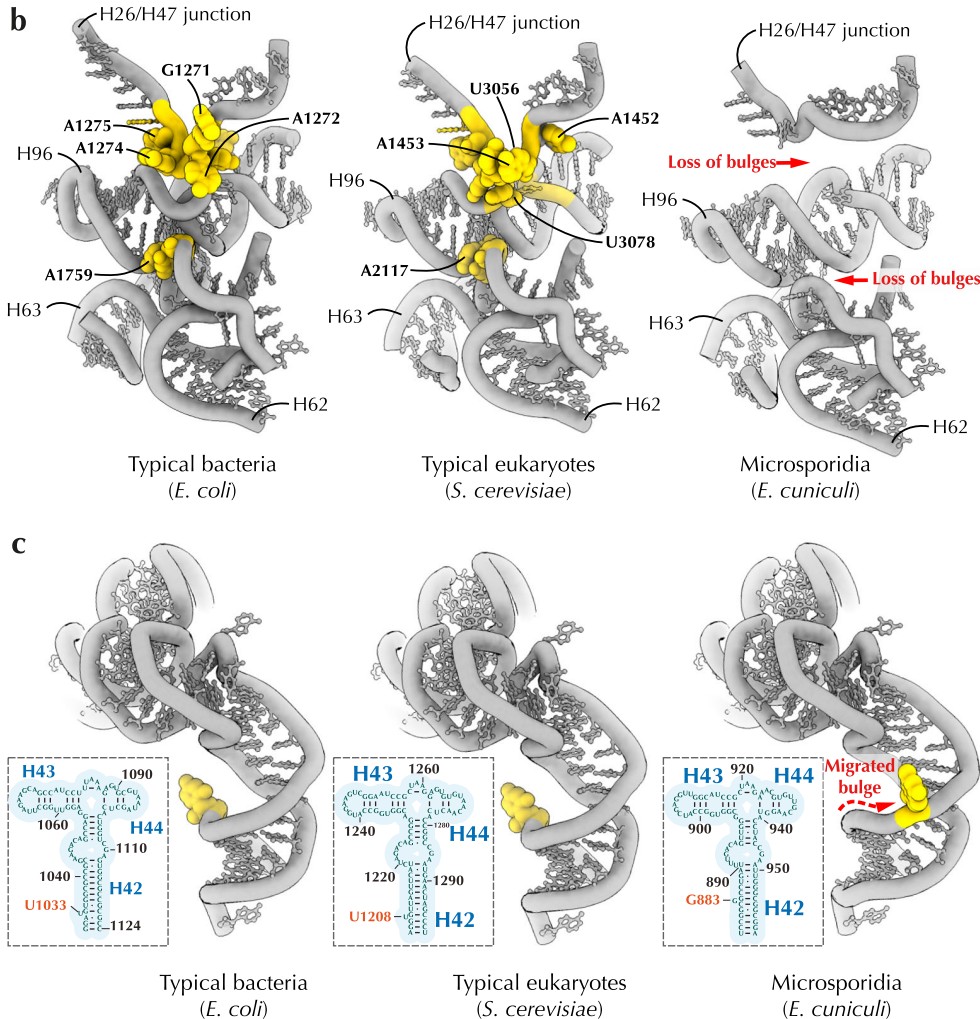

**Fig. 4 Microsporidian rRNA is stripped of highly conserved bulges. a** rRNA bulges (*S. cerevisiae* numbering) that are missing in the structure of *E. cuniculi* ribosomes but present in most other eukaryotes **b** Comparison of the ribosome interior of *E. coli*, *S. cerevisiae*, *H. sapiens* and *E. cuniculi* illustrates that microsporidian parasites lack many ancient, highly conserved rRNA bulges. These bulges stabilize ribosome structure, therefore their absence in microsporidia suggests decreased stability of rRNA folding in microsporidian parasites. **c** Comparison with the P-stalk (L7/L12-stalk in bacteria) illustrates that the loss of rRNA bulges can occasionally co-occur with the emergence of new bulges in the vicinity of the lost ones. The helix H42 in 23 S/28 S rRNA possesses an ancient bulge (U1206 in *S. cerevisiae*), which is estimated to be at least 3.5 billion years old due to its conservation across the three domains of life. In microsporidia, this bulge has been eliminated; however, a new bulge (A1306 in *E. cuniculi*) has evolved in the vicinity of the lost one.

*Ordospora* species but is absent in other species, including other microsporidians. In the ribosome structure, msL2 occupies a void formed by the loss of the rRNA expansion ES31L. In this void, msL2 helps stabilize rRNA folding and likely compensates for the ES31L loss (Fig. 6).

We next compared msL2 protein with the previously described protein msL1—the only known microsporidia-specific ribosomal protein that was found in *V. necatrix* ribosomes[31]. We wanted to test whether msL1 and msL2 are evolutionary related to each other. Our analysis showed that msL1 and msL2 occupy the same cavity in the ribosome structure, but have distinct primary and tertiary structure, suggesting their independent evolutionary origin (Fig. 6). Thus, our finding of msL2 provided evidence that close groups of eukaryotic species can independently evolve structurally distinct

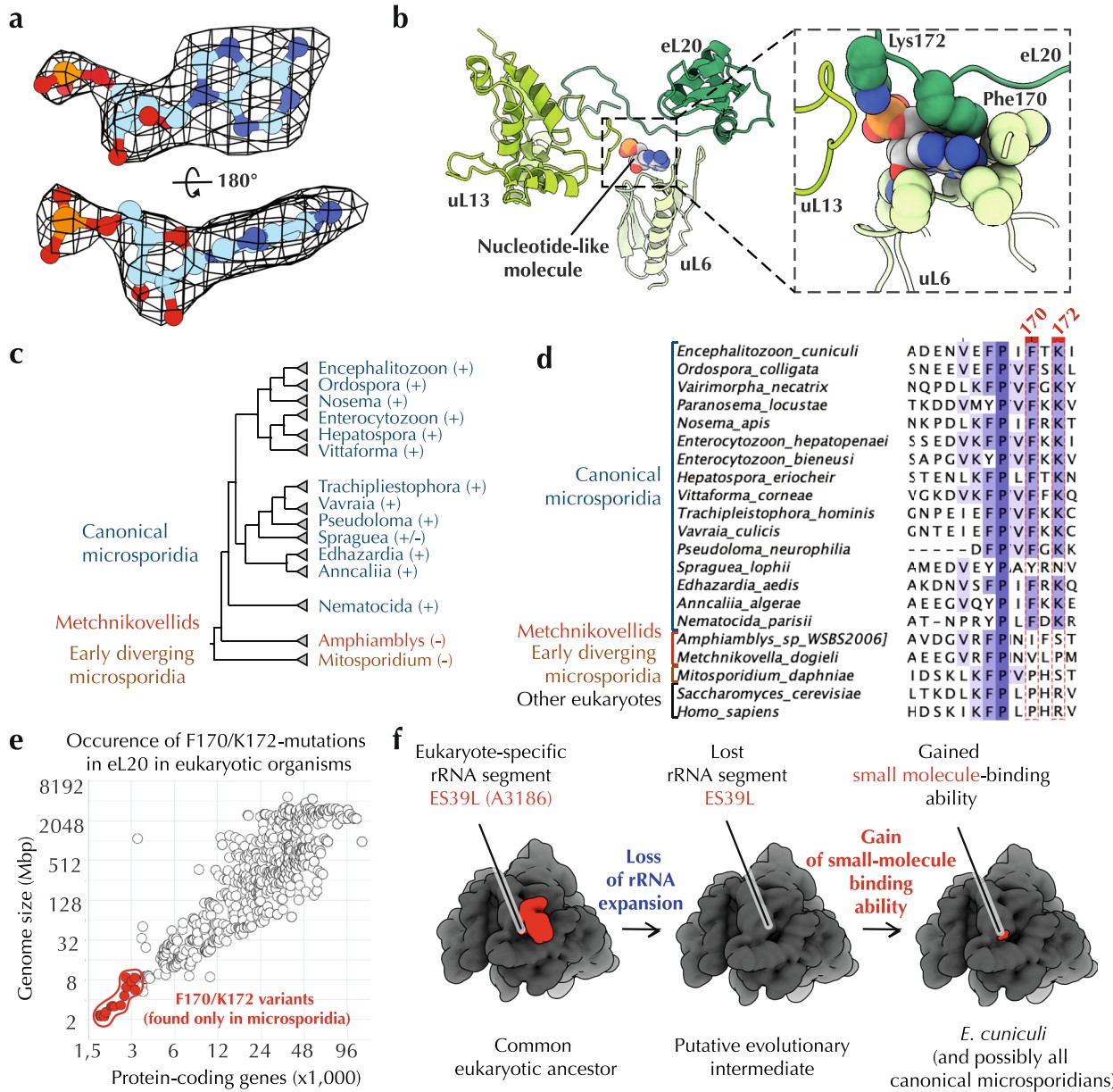

**Fig. 5 Microsporidian ribosomes use small molecules as ribosomal building blocks to compensate for the rRNA and protein reduction. a** The cryo-EM map indicates the presence of the extra-ribosomal nucleotide bound to the *E. cuniculi* ribosome. In the *E. cuniculi* ribosome this nucleotide occupies the same space as the 25 S rRNA nucleotide A3186 (*S. cerevisiae* numbering) in most other eukaryotic ribosomes. **b** In the *E. cuniculi* ribosome structure, this nucleotide is being sandwiched between ribosomal proteins uL9 and eL20, stabilizing contacts between these two proteins. **c–d** Analyses of eL20 sequence conservation in microsporidian species. A phylogenetic tree of microsporidian species (**c**) and a multiple sequence alignment of protein eL20 (**d**) illustrate that the nucleotide-binding residues F170 and K172 are conserved in most canonical microsporidia (aside from *S. lophii*), except for the early-branched microsporidia, in which the rRNA expansion ES39L is preserved. **e** The plot shows that the nucleotide-binding residues F170 and K172 are only found in eL20 from microsporidian parasites with highly reduced genomes and not in other eukaryotes. Overall, these data indicate that microsporidian ribosomes have evolved a nucleotide-binding site that appears to bind AMP molecules and use them to stabilize protein–protein interactions in the ribosome structure. The high degree of conservation of this binding site among microsporidia and its absence in other eukaryotes indicates that this site may provide a selective advantage for microsporidia survival. Therefore, the nucleotide-binding pocket in microsporidian ribosomes appears not to be a vestigial feature or the ultimate form of rRNA degeneration, as previously suggested[32], but a useful evolutionary innovation that allows microsporidian ribosomes to directly bind small molecules, utilizing them as molecular building blocks for ribosome assembly. This finding makes microsporidian ribosomes the only known ribosomes that use single nucleotides as a structural building block. **f** A hypothetic evolutionary path of the nucleotide-binding acquisition.

ribosomal proteins to compensate for the loss of rRNA segments. This finding is remarkable because most cytoplasmic eukaryotic ribosomes have invariant protein content, comprising the same set of 81 families of ribosomal proteins[47]. The birth of msL1 and msL2 in distinct microsporidian branches in response to the loss of the rRNA expansion segments suggests that degeneration of parasitic

molecular structures forces parasites to seek compensatory mutations that may eventually lead to gain of compositional diversity of these structures in distinct groups of parasites.

**Sequence hypervariability of ribosomal proteins.** Finally, when our model building was complete, we compared the composition of

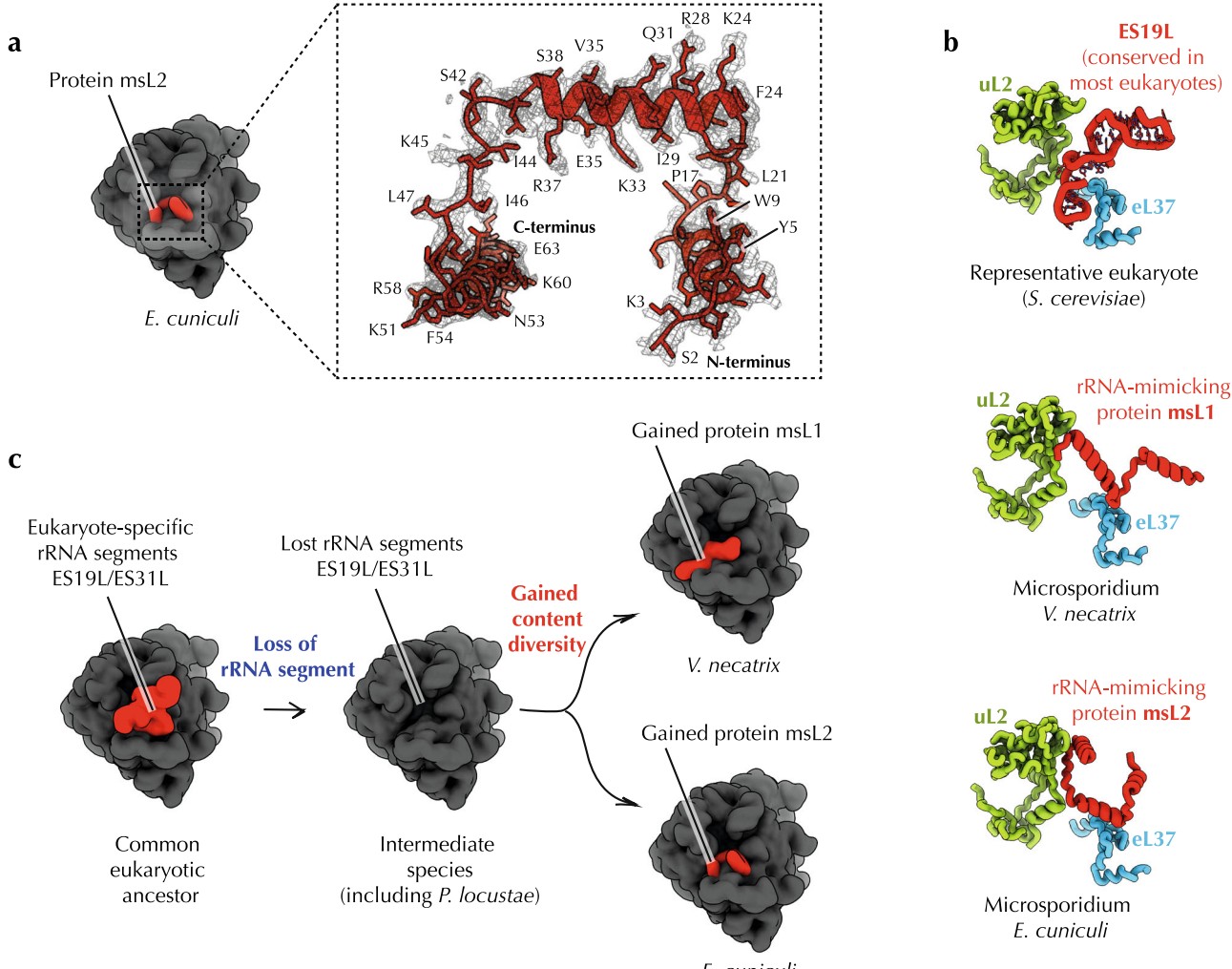

**Fig. 6 The loss of rRNA expansions leads to diversification of protein content in microsporidian ribosomes. a** Electron density and model of the microsporidia-specific ribosomal protein msL2 found in *E. cuniculi* ribosomes. **b** Most eukaryotic ribosomes, including 80 S ribosomes of *S. cerevisiae*, possess the rRNA expansion ES19L, which has been lost in most microsporidian species. The previously determined structure of microsporidian ribosomes from *V. necatrix* showed that the loss of ES19L in these parasites was compensated by the evolution of a new ribosomal protein, msL1. In this study, we discovered that *E. cuniculi* ribosomes also evolved an additional RNA-mimicking ribosomal protein as an apparent compensation for the loss of ES19L. However, msL2 (currently annotated as hypothetical protein ECU06_1135) and msL1 have different structure and evolutionary origin. **c** This finding of the birth of evolutionary unrelated ribosomal proteins msL1 and msL2 illustrates that ribosomes can achieve an unprecedented level of compositional diversity, even within a small group of closely related species, if they accumulate a deleterious mutation in their rRNA. This finding may help shed light on the origin and evolution of mitochondrial ribosomes, which are known for their severely reduced rRNA and exceptional variability of protein composition among species.

*E. cuniculi* ribosomes with the composition that was predicted based on genome sequence[27]. Previously, the *E. cuniculi* genome was predicted to lack several ribosomal proteins, including eL14, eL38, eL41, and eS30 due to the apparent absence of their homologs in the *E. cuniculi* genome[27,48]. The loss of multiple ribosomal proteins was also predicted in most other intracellular parasites and endosymbionts with highly reduced genomes[49]. For example, while most free-living bacteria contain the same set of 54 families of ribosomal proteins, only 11 of these protein families have detectable homologs in each of the analyzed genomes of host-restricted bacteria[49]. Supporting this idea, the loss of ribosomal proteins was observed experimentally in microsporidians *V. necatrix* and *P. locustae*, which both lack proteins eL38 and eL41[31,32].

Our structure revealed, however, that only eL38, eL41, and eS30 are in fact lost in *E. cuniculi* ribosomes. The protein eL14 was retained, and our structure revealed why this protein could not be detected through homology search (Fig. 7). In *E. cuniculi*

ribosomes, most of the eL14-binding site is lost due to degeneration of the rRNA expansion ES39L. In the absence of ES39L, eL14 loses most of its secondary structure, and only 18% of the eL14 sequence is identical between *E. cuniculi* and *S. cerevisiae*. This poor sequence conservation is remarkable because even *S. cerevisiae* and *H. sapiens*—organisms that are separated by 1.5 billion years of evolution—possess more than 51% identical residues in eL14. This extraordinary loss of conservation explains why *E. cuniculi* eL14 is currently annotated as hypothetical protein M970_061160 rather than ribosomal protein eL14[27].

This finding illustrates that rRNA degeneration can lead to drastic loss of sequence conservation in adjacent ribosomal proteins, rendering these proteins undetectable for homology search. Hence, we may overestimate the actual extent of molecular degeneration in organisms with small genomes because some proteins that are viewed as lost are in fact preserved, though in a highly altered form.

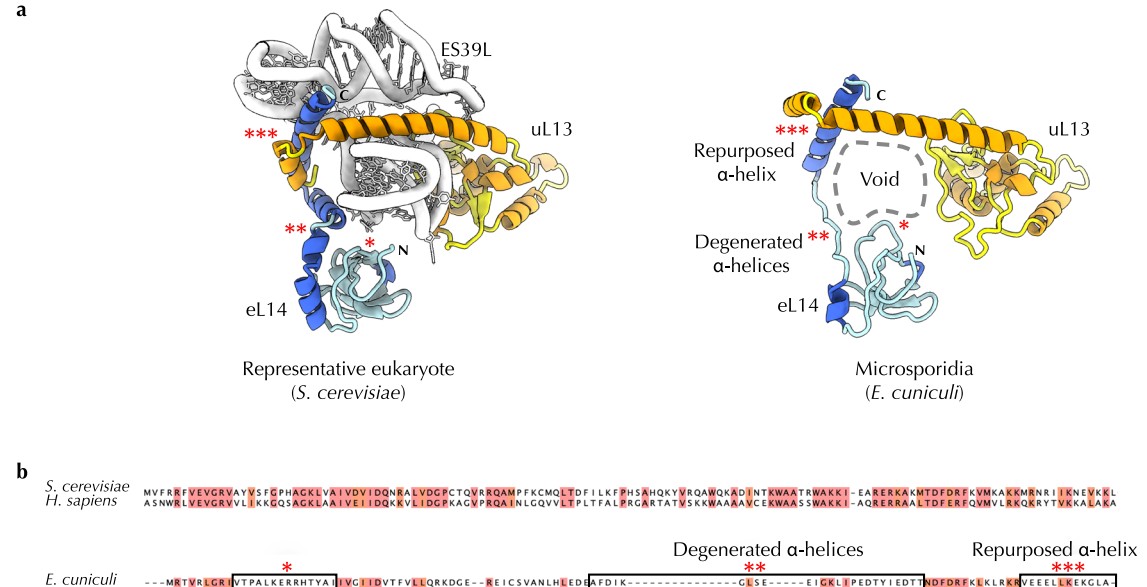

**Fig. 7 Hypervariable sequences can render microsporidian proteins undetectable for homology searches. a** Microsporidian ribosomes have lost the rRNA expansion ES39L, which partially eliminated the binding site for ribosomal protein eL14. In the absence of ES39L, microsporidian protein eL14 underwent a loss of secondary structure, in which former rRNA-binding α-helices degenerated to minimal-length loops. **b** Multiple sequence alignment shows that protein eL14 is highly conserved among eukaryotic species (where it shares 57% sequence identity between yeast and human homologs), but poorly conserved and divergent among microsporidia (where no more than 24% of residues are identical to eL14 homologs from *S. cerevisiae* or *H. sapiens*). This poor sequence conservation, along with changes in the secondary structure, explains why homologs of eL14 have never been found in *E. cuniculi* and why this protein was thought to have been lost in *E. cuniculi*. Instead, *E. cuniculi* eL14 was previously annotated as the hypothetical protein M970_061160. This observation reveals that the diversity of microsporidian genomes is currently overestimated: some genes that are currently thought to have been lost in microsporidia are actually retained, although in a highly diverged form; conversely, some genes that are thought to encode microsporidia-specific proteins (e.g., hypothetical protein M970_061160) do in fact encode highly divergent proteins that can be found in other eukaryotes.

## Discussion

How do parasites retain the function of their molecular machines in the face of extreme genome reduction? Our study provides answers to this question by describing a complex molecular structure—the ribosome—from *E. cuniculi*, an organism with one of the smallest eukaryotic genomes.

For nearly two decades, proteins and RNA molecules in microbial parasites have been known to often differ from homologous molecules in free-living species by lacking quality control centres, being reduced to up to 50% of their size in free-living microorganisms and having numerous debilitating mutations that compromise folding and function. For instance, ribosomes from organisms with small genomes, including many intracellular parasites and endosymbionts, were predicted to lack multiple ribosomal proteins and up to a third of rRNA nucleotides compared to free-living species[27,29,30,49]. How these molecules are nonetheless functional in parasites remains largely a riddle studied primarily by means of comparative genomics.

Our study shows that the structures of macromolecules can reveal many aspects of evolution that are hard to extract from traditional comparative genomics studies of intracellular parasites and other host-restricted organisms (Supplementary Fig. 7). For instance, the example of protein eL14 illustrates that we can overestimate the actual extent of degeneration of molecular machineries in parasitic species. Currently, *Encephalitozoon* parasites are thought to possess hundreds of microsporidia-specific genes; however, our findings suggest that some of these seemingly specific genes are in fact only highly divergent variants of genes that are widely present in other eukaryotes. In addition, the example of protein msL2 illustrates how we can overlook new ribosomal proteins and underestimate the contents of parasitic molecular machineries. And the example of small molecules shows how we can overlook the most ingenious innovations in parasitic molecular structures that may endow them with new biological activities.

Collectively, these findings improve our understanding of the differences between molecular structures of host-restricted organisms and their counterparts in free-living organisms. We show that the molecular machineries long viewed as reduced, degenerated, and suffering from various debilitating mutations instead possess an array of systematically overlooked and extraordinary structural features.

On a separate note, our finding of bulgeless and molten rRNA segments in *E. cuniculi* ribosomes illustrates that genome reduction can alter even those parts of essential molecular machineries that are conserved across the three domains of life—in species that are separated by almost 3.5 billion years of independent evolution.

Bulgeless and molten rRNA segments in *E. cuniculi* ribosomes are particularly interesting in the light of previous studies of RNA molecules from endosymbiotic bacteria. For example, in the endosymbiont of aphids *Buchnera aphidicola*, rRNA and tRNA molecules were shown to have heat-sensitive structures due to the A + T compositional bias and the high proportion of non-canonical base pairs[20,50]. These changes in RNA, along with changes in protein molecules, are currently thought to underlie the excessive reliance of endosymbionts on chaperons and the inability of endosymbionts to tolerate heat[21,23]. Although microsporidian parasites possess structurally distinct changes in their rRNA, the nature of these changes suggests that reduced thermostability and higher dependance on chaperones might be a commonly occurring feature in RNA molecules of organisms with reduced genomes.

On a separate note, our structure reveals that microsporidian parasites have evolved the unique ability to withstand degeneration of widely conserved molecular structures by evolving the

capacity to use abundant and readily available small metabolites as structural mimics of degenerated rRNA and protein segments. This idea is supported by the fact that the small molecules, which compensate the loss of rRNA and protein segments in *E. cuniculi* ribosomes, are bound to microsporidia-specific residues in proteins uL15 and eL30. This suggests that small molecules' binding to the ribosome is likely a product of positive selection in which microsporidia-specific mutations in ribosomal proteins were selected for their ability to increase the ribosome affinity to the small molecules, possibly leading to more effective ribosome biogenesis. This finding uncovers one ingenious innovation in molecular structures of microbial parasites, providing us with a better understanding of how parasitic molecular structures can retain their function despite their reductive evolution.

Currently, the identify of these small molecules remains unclear. It is also unclear why the occurrence of these small molecules in the ribosome structure differs among microsporidian species. Specifically, it is unclear why the nucleotide binding was observed in *E. cuniculi* and *P. locustae* ribosomes but not in *V. necatrix* ribosomes, despite the presence of residues F170 and K172 in *V. necatrix* protein eL20. This absence may be caused by residue 43 of uL6 (located near the nucleotide-binding pocket), which is a tyrosine in *V. necatrix*, rather than threonine, as in *E. cuniculi* and *P. locustae*. It is possible that the bulky aromatic sidechain of Tyr43 prevents nucleotide binding due to steric overlap. Alternatively, the apparent nucleotide absence could be caused by the lower resolution of the cryo-EM map, precluding modeling of that segment of *V. necatrix* ribosomes.

On a separate note, our work illustrates that the process of genome decay may act as a force of invention. Specifically, the structure of *E. cuniculi* ribosomes indicates that the loss of rRNA and protein segments in microsporidian ribosomes creates the evolutionary pressure to facilitate structural variations in the ribosome structure. Occurring far from ribosomal active sites, these variations appear to help preserve (or restore) the optimal ribosome assembly that would be otherwise compromised by the rRNA reduction. This suggests that major innovations in microsporidian ribosomes appear to have evolved as a need to buffer genetic drifts.

This is perhaps best illustrated by the nucleotide binding, which has never been observed in other organisms—so far. The fact that the nucleotide-binding residues are present in canonical microsporidia but not in other eukaryotes indicates that the nucleotide-binding site is not a mere vestige awaiting extinction or an ultimate form of rRNA reduction to a single nucleotide[32]. Instead, this site appears to be a useful feature that likely evolved through multiple rounds of positive selection. It is possible that the nucleotide-binding site emerged as a byproduct of natural selection: once ES39[L] was degenerated, microsporidia were forced to seek a compensation to restore optimal ribosome biogenesis in the absence of ES39[L]. And because the nucleotide could mimic the molecular contacts of the nucleotide A3186 in ES39[L], a nucleotide molecule became a ribosomal building block, with its binding further improved by mutations in the eL30 sequence.

Regarding the molecular evolution of intracellular parasites, our study indicates that the forces of the Darwinian natural selection and genetic drifts of genome decay do not act in parallel but rather act in a swing-like fashion. First, genetic drifts eliminate important features in biological molecules, creating a pressing need for a compensation. And only then, while parasites are fulfilling this need through Darwinian natural selection, their macromolecules gain a chance to evolve their most spectacular and innovative features. Importantly, the evolution of the nucleotide-binding site in *E. cuniculi* ribosomes suggests that this lose-to-gain pattern of molecular evolution not only buffers deleterious mutations but occasionally endows parasitic macromolecules with radically new functions.

This idea is consistent with Sewall Wright's shifting balance theory, which states that a strict regime of natural selection restricts organisms' ability to innovate[51–53]. However, if natural selection is violated by genetic drifts, these drifts can create changes that are themselves not adaptive (or even deleterious) but lead to further changes that give higher fitness or new biological activities. Our structure supports this idea, illustrating that the very same types of mutations that degrade the folding and function of biological molecules appear to serve as the major trigger of their improvement. Consistent with a lose-to-gain pattern of evolution, our study suggests that genome decay, traditionally viewed as a process of degeneration, is also a major engine of innovation that can occasionally, perhaps even often, allow macromolecules to acquire new activities that parasites may use to their advantage.

## Methods

**Producing microsporidian spores**. *E. cuniculi* strain II[54] was grown in co-culture with RK-13 cells (CCL37; American Type Culture Collection, Manassas, Va.) passaged in minimal essential medium (MEM) containing 10 % heat-inactivated-foetal calf serum and antibiotics (penicillin/streptomycin (0.1 mg/ml), ampicillin B (1 µg/ml) and kanamycin sulphate (0.1 mg/ml))[55]. For the experiment, 14 ×175 cm² flasks of infected RK-13 cells were harvested. The infected cells were scraped from each flask in PBS buffer containing 0.1% Triton X-100, passed 5 times through a 27 G needle, pelleted by centrifugation at 4 °C and 3000 g for 5 min, and resuspended in 10 ml of PBS buffer containing 0.1% Triton X-100. To release the spores from the host cells, the suspension was placed on ice and sonicated three times for 1 min per round, with 1 min interruption on ice between each sonication. To separate the spores from the host cell debris, the sonicated suspension was carefully layered on top of a 25% Percoll solution. The spores were then pelleted by centrifugation at 4 °C and 900 g for 30 min[56]. The pellet was washed three times with PBS and pelleted using centrifugation at 4 °C and 3000 g for 5 min after each wash. The resulting pellet containing *E. cuniculi* purified spores had a weight of ~300 mg.

**Isolating *E. cuniculi* ribosomes**. To purify ribosomes, ~300 mg of spores were resuspended in 300 µl of buffer O (HEPES-KOH pH 7.5, 25 mM MgCl₂, 130 mM KCl) that was supplemented with 1% (w/v) PEG 20,000. The spore suspension was transferred into a BeadBug™ 2.0 ml tube prefilled with 0.5 mm Zirconium beads, and the spores were disrupted by using Savant Bio101 Fp120 cell disrupter (Thermo) with one cycle of bead beating at room temperature that lasted for 45 s at 6.5 relative shaking speed. The lysate was cleared by centrifugation at 4 °C and 16,000 g for 4 min. The supernatant was transferred into a new 2 ml tube and mixed with 25% (w/v) solution of PEG 20,000 in water to increase PEG 20,000 concentration from 1% to 4% to precipitate ribosomes. The solution was then incubated for 5 min on ice, and ribosomes were pelleted by centrifugation at 4 °C and 16,000 g for 4 min. The pellet with a crude ribosome sample was dissolved in ~70 µl of buffer O, and 20 µl of this solution were used for cryo-EM analysis (Supplementary Fig. 1).

**Preparation of grids and data collection**. Quantifoil grids (R1.2/1.3, 400 mesh, copper) were glow discharged (10 mA, 30 s, Quorum GloQube), and 3 µL of the crude sample of *E. cuniculi* ribosomes (~300 nM) was pipetted onto a grid. The excess sample was immediately blotted off and vitrification was performed by plunging the grid into liquid nitrogen-cooled liquid ethane at 100% humidity and 4 °C using an FEI Vitrobot Mark IV (Thermo Fisher). The data were collected on a Thermo Fisher Titan Krios electron microscope (Astbury Biostructure Laboratory, University of Leeds, UK) at 300 kV. Similar to our previous study[57], data collection was set up by exposing the sample to an electron dose of 60 e⁻/Å² across 1.35 s. 2210 micrograph movies were recorded by a Thermo Fisher Falcon 3EC detector in integrating mode, split into frames which each received a dose of 1.14 e⁻/Å². A nominal magnification of 96,000x was applied, resulting in a final object sampling of 0.861 Å/pixel. A defocus range of −0.8 to −2.6 µm was used (Supplementary Table 1).

**Image processing and map generation**. Drift-corrected and dose-corrected averages of each movie were created using RELION 3.1's own implementation of motion correction and the contrast transfer functions estimated using CTFFIND-4.1[58]. All subsequent image processing steps were carried out using RELION 3.1[59]. Particles were picked using Laplacian-of-Gaussian autopicking and reference-free 2D classification used to generate templates for further autopicking. The resulting particles were extracted with binning-by-4, and 2D and 3D classification performed to remove junk images. The 3D classification step was carried out using a 60 Å low-passed filtered ab initio starting model made using a stochastic gradient descent procedure. The remaining particles were re-extracted without binning and aligned and refined in 3D, again using a 60 Å low-passed filtered ab initio starting model.

Rounds of CTF refinement and Bayesian polishing were performed until the map resolution stopped improving. 108,005 particles fed into the final 3D reconstruction of estimated resolution 2.7 Å.

Multibody refinement was performed using soft extended masks to define the 60S, 40S body and 40S head as rigid bodies (Supplementary Fig. 2). This procedure used iteratively improved partial signal subtraction and focussed refinement to generate reconstructions for each body at estimated resolutions of 2.6, 2.9 and 3.1 Å, respectively[60]. All resulting reconstructions were subjected to post-processing in RELION, which included solvent masking, estimation and correction for B-factors, and low-pass filtering according to local resolution (estimated using RELION's own implementation). Sharpened maps were also produced using DeepEMhancer[61]. DeepEMhancer maps were used to aid initial model building, whereas maps filtered by local resolution within RELION were used for both model building and model refinement. In all cases, final resolutions were estimated using the gold-standard Fourier shell correlation (FSC = 0.143) criterion (Supplementary Fig. 2).

**Model building and refinement.** The model was built using fragments of *S. cerevisiae* (pdb id 4v88) and *V. necatrix* ribosomes (pdb id 6rm3) as starting models that were edited using Coot[62] using genomic sequences of the *E. cuniculi* strain GB-M1 to model rRNA and ribosomal proteins. For ribosomal proteins that are encoded by two alternative genes (with one gene coding for a zinc-coordinating protein and another gene coding for a zinc-free ribosomal protein), we used zinc-coordinating isoforms, because the cryo-EM map revealed the presence of these isoforms and not their zinc-free paralogs in the ribosome structure. The identity of protein msL2 in the ribosome structure was determined using the genomic sequence of the *E. cuniculi* strain GB-M1 and the cryo-EM map that revealed a unique combination of aromatic and bulky amino acids in its structure: the cryo-EM map showed that msL2 has a tyrosine residue at position 5, a tryptophan residue at position 9, and lysine or arginine residues at positions 10, 12, and 13. The only protein with this sequence was the hypothetical protein ECU06_1135, whose sequence and length were fully consistent with the cryo-EM map.

The structure of *E. cuniculi* ribosomes was refined using Phenix real space refine[63] and validated using MolProbity[64] within Phenix and PDB OneDep[65]. The parts of the model corresponding to the 60S, 40S body and 40S head were built and refined using the consensus map, 40S body multibody map and 40S head multibody map, respectively (Supplementary Fig. 2). Details of the final model are found in (Supplementary Table 1). Model refinement and validation statistics are found in (Supplementary Table 1). Figures were produced using UCSF ChimeraX[66] and Adobe Illustrator.

**Quantification and statistical analysis.** All cryo-EM data sets were processed using RELION (Supplementary Table 1). All resolutions reported are based on the gold-standard FSC 0.143 criterion (Supplementary Fig. 2). FSC curves were calculated using soft-edged masks. Refinement statistics of all molecular models are summarized in (Supplementary Table 1). These models were also evaluated based on MolProbity scores[64] and Ramachandran plots.

**Analysis of ribosomal proteins.** Sequences of eukaryotic ribosomal proteins were retrieved from Uniprot (https://www.uniprot.org/) using a custom python script whose code is available in (Supplementary Data 1). The list of eukaryotic organisms with fully sequenced genomes was obtained from NCBI (https://www.ncbi.nlm.nih.gov/genome/browse#!/overview/). The retrieved sequences are available in (Supplementary Data 2, 3), and their alignments were generated using Clustal Omega with default settings.

**Reporting Summary.** Further information on research design is available in the Nature Research Reporting Summary linked to this article.

## Data availability

Data that support the finding of this study have been deposited in the EMDB/PDB databank with the accession code pdb id 7QEP [https://doi.org/10.2210/pdb7QEP/pdb] for the structure of the *E. cuniculi* ribosome, and EMD-13936 for the corresponding cryo-EM maps. The code of our custom python script to retrieve sequences of ribosomal proteins is available in Supplementary Data 1.

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

## Acknowledgements

We thank members of the Melnikov, Ranson and Hirt laboratories, along with Martin Embley, Jeff Errington, Neil Perkins, and Tracy Palmer (all Newcastle University, UK), Alexander Mankin (University of Illinois at Chicago, USA), and Bernhard Kuhle (Scripps, USA) for their feedback and help with preparing this manuscript. We also thank Elizabeth S. Didier (Tulane National Primate Research Center, USA) for providing the *E. cuniculi* strain. This work was supported by the Wellcome Trust (4-year PhD studentship 203743/Z/16/Z to D.N.), the Medical Research Council (G0901526 and MR/N009738/1 to M.S.), and the European Union's Horizon 2020 research and innovation programme (the Marie Skłodowska-Curie Grant Agreements No. 97617 to J.P. and No. 895166 to S.M). All cryo-EM was performed at the Astbury Biostructure Laboratory (ABSL) funded by the University of Leeds and the Wellcome Trust (108466/Z/15/Z). We thank ABSL facility staff for their help with cryo-EM data collection.

## Author contributions

R.P.H., N.A.R., and S.V.M. designed the study; J.P. purified *E. cuniculi* spores; S.V.M. purified *E. cuniculi* ribosomes; D.N. collected cryo-EM data, produced cryo-EM maps and refined the atomic model; M.S., K.H-B. and S.V.M. built the atomic model of the microsporidian ribosome; C.R.B. and S.V.M. identified the microsporidia-specific protein msL2; S.V.M. and C.R.B. performed the evolutionary analyses; S.V.M. wrote the manuscript with input and feedback from all the co-authors.

## Competing interests

The authors declare no competing interests.
