## [Peer Review File · Nature Communications]

Adaptation to genome decay in the structure of the smallest eukaryotic ribosomesReviewers' Comments:

Reviewer #1:

Remarks to the Author:

The manuscript by Nicholson et al. reports a structural investigation aimed at characterizing one of the smallest known cytosolic eukaryotic ribosomes from the parasitic microorganism *Encephalitozoon cuniculi* possessing one of the smallest and most reduced eukaryotic genomes in nature. This study follows up on two similar studies from the Klinge (Nature microbiology 2019) and the Barandun (PLOS Biology 2020) groups describing the structures of the cytosolic ribosome from two other species of microsporidians, *Paranosema locustae* and *Vairimorpha necatrix*. From an evolutionary view point, the study is certainly interesting and reports several novel observations regarding the molecular adaptation of the ribosomes from these peculiar organisms, such as the utilization of small molecules to mimic and compensate for degeneration of various protein and rRNA features. Other striking observation is the persistence of bulgeless rRNA segments and the development of "molten rRNA linkers" that link various rRNA segments through loosely pairing single strands instead of regular rRNA helices. The authors coin the expression "lose-to-gain evolution" to characterize this revealed pattern of molecular adaptation/evolution.

The manuscript is well written and the figures are clear, in addition of being aesthetic. The references are sufficient and methods are appropriately detailed. I have no major concerns and warmly recommend the publication of this elegant study. I only have minor comments:

The authors should provide model vs. map curves as well, in order to appreciate the model fitting with the nicely detailed maps of the study.

To fully fathom the reduction of the rRNA and to compare it with other reduced rRNAs and to yeast rRNA that is used as a reference, it is capital to provide a 2D structure diagram of the *E. cuniculi* rRNA! Ideally side-by-side with *P. locustae* or *V. necatrix* and yeast.

Figure S2: Local resolution panels show the resolution/color scale with probably disordered numbers (2.3 – 2.9 – 2.6 Å), perhaps the correct order is 2.3 – 2.6 – 2.9 Å?

The authors discuss the existence of various small exo-ribosomal molecules such as a possible spermidine and they illustrate it by Figure S7. Although it is rather clear, perhaps the authors would want to consider some transparency in the mesh of the lower panels and some smoothing (increased sampling/number of mesh), as it stands the mesh is quite opaque. This is a purely aesthetic comment.

In the discussion, the authors compare the evolution of mitoribosomes to cytosolic ribosomes of microsporidians, stating that they followed similar evolutionary paths mainly regarding the rRNA reduction. I don't agree with this conclusion, as the diversity of mitoribosomes clearly showed diverse ways of evolutions. Indeed, some mitoribosomes have evolved to gain more rRNA such as yeast and more drastically, plants very widely. Indeed, plants mitoribosomes count nearly 20 to 30% more rRNA mass compared to bacteria. Other mitoribosomes are on the other extreme of the spectra and have indeed lost a significant mass of rRNA, such as mammalian mitoribosomes and the much more extreme case of kinetoplastids, especially in the small ribosomal subunit. And to make things even more complicated, some mitoribosomes have evolved to be assembled from truncated pieces of rRNA, like in many species of algae where a dozen specific mitoribosomal proteins appeared to "glue" the different pieces of rRNA. Finally, in all these mitoribosomes, the acquisition of numerous species-specific proteins is characteristic, several folds more novel proteins compared to microsporidia cytosolic ribosomes. It is believed that the evolution of mitoribosomes commenced with an RNA-constructive phase where more RNA was acquired, thus calling for the acquisition of more proteins to stabilize and accompany the extra mass of RNA, followed by a slow and progressive RNA-destructive phase where in part under the pressure of more compact mitochondrial genomes, rRNA decreases leaving the acquired proteins to ensure structural support and regulatory functions that once were

performed by rRNA. I simply recommend the authors to remove this part of their discussion, or at least reduce it to the simple comparison with some mitoribosomes.

Reviewer #2:

Remarks to the Author:

Intracellular pathogens often undergo genomic reduction, but the molecular consequences of this reductive evolution are not well understood. The microsporidian *Encephalitozoon cuniculi* is a drastically reduced intracellular parasite, having one of the smallest known eukaryotic genomes, making this species an excellent system to study the effect genomic reduction on protein function. In this manuscript, the authors characterize and analyze a high-resolution cryo-EM structure of the *E. cuniculi* ribosome. Several microsporidian ribosomal structures have been examined by cryo-EM, but *E. cuniculi* is the most reduced of these and analysis of this ribosomal structure revealed numerous structural adaptations. The authors describe these novel features including loss of conserved RNA bulges, the formation of truncated and linearized rRNA segments to minimize the length of rRNA bridges, and the binding of small molecules. Additionally, the authors identified a novel ribosomal protein and found that eL14, which was previously thought to be lost, was retained but hard to detect by sequencing homology. Together, this is a very interesting and thorough study that describes innovative features that occur in an extremely derived ribosome. Although biochemical testing of the features observed from the structure would strengthen the study, the high resolution of the structure and the comparison to other structures allows these observations to stand on their own.

Main points:

1. It is interesting that nucleotide binding was observed in *E. cuniculi* and *P. locustae*, but not in *V. necatrix*. Why is that? Are the F170 and K172 residues not present in *V. necatrix* or is there some other explanation?
2. For Figure 4D, it would be useful to add the sequences of *P. locustae* and *V. necatrix*. It would also be useful to add a Nematocida species. Labeling amphitrypan and mitosporidium as they are in C would be more informative than labeling them "other eukaryotes". The figure legend states that "illustrate that the nucleotide-binding residues F170 623 and K172 are conserved across all microsporidian species, except for the early-branched microsporidia", but this appears to not be the case as *S. lophii* does not contain these residues.
3. On line 215, it is mentioned that the Asp51 and Arg56 mutations in uL15 are microsporidia-specific. However, in figure S7, these mutations are only shown for *E. cuniculi*. It is unclear whether these mutations are conserved amongst microsporidia, or only occur in *E. cuniculi*.

Minor point:

1. The dormancy factor MDF1 is mentioned in figure 1 and figure S5, but not in the main text. It would be good to add to the main text description of the results related to MDF1.

Reviewer #3:

Remarks to the Author:

In the presented work by Nicholson et al., the authors reported a new cryo-EM structure of the functionally inactive dormant ribosome from one of the smallest eukaryotes *Encephalitozoon cuniculi*, which belongs to the microsporidia group. This is not the first structure of the microsporidian ribosome and definitely not the first structure of the eukaryotic ribosome. However, in comparison to the

previously reported two structures of ribosomes from microsporidia, this one appears to have the highest resolution reported to date. Moreover, given that the ribosomes from this species of microsporidia underwent one of the largest known reductions of size, this structure is impactful from a fundamental point of view because it can shed light on the evolution of ribosomes in this group of intracellular parasites of eukaryotes.

Overall, this work is presented as a very concise and clearly written manuscript with beautiful and stylish illustrations that definitely capture an eye. The strong side of this work is the better structural model of the microsporidian ribosome, which in addition are probably very hard to isolate and purify. However, there are several major concerns denoted below that need to be addressed by the authors.

Major issues:

1. Authors claim that they identified AMP and spermidine-like molecules stably bound to the microsporidian ribosome. I do agree that the cryoEM map that is shown in Figure 4A does resemble AMP in shape, but the structural methods (unless we are talking about atomic resolution structures at 1-1.4Å) are not the methods typically used to identify the molecules. The authors should provide mass-spec data or some other biochemical data to add more credibility to their assignments of AMP.

2. Regardless of whether that extra density corresponds to AMP or not, typically, it is OK to speculate and make assumptions. However, building new assumptions on assumptions is not a good practice. In other words, what if that density is not an AMP? Why couldn't it be a GMP molecule, for example? What if this is something else? Then all the discussions about its new role in sustaining the structure of the microsporidian ribosome do not have sense anymore. It is very dangerous to make such second-level assumptions without providing compelling experimental evidence.

3. Although the manuscript indeed is nicely written, the style appeared somewhat more like a review rather than an experimental paper. I think the authors should describe in more detail the peculiarities of ribosome purification and structure determination rather than speculating about possible evolution on every page. This work is abundant with comparisons of various small structural features of the ribosome, which is important (no doubt) but is missing comparisons of the global structures of the ribosomes from different groups of organisms with the current one.

Minor issues:

4. At the very beginning of the results section, the authors should elaborate more on how ribosomes were purified from the spores. I guess it is not really a trivial task to purify ribosomes from microsporidian spores, and, therefore, it is likely to be interesting to the reader. Moreover, this is definitely not something that is routinely done in the labs, and therefore it is important to emphasize how difficult it could be. Yes, of course, the details of it are in the methods, but the story will likely benefit from adding a few sentences explaining how purification was done and what are the typical pitfalls of this procedure. I would also recommend adding a few sentences here explaining how the structure was determined, how particles were classified, what masks were applied to which parts of the ribosome, etc.

5. Because the ribosomes were isolated from the spores, apparently, they represent a dormant, inactive state, which is mentioned in a couple of places. I think this is a very important aspect that this structure does not represent a functionally active state of the ribosome from microsporidia and should be explicitly stated in the main text.

6. Figure 1: Panel A is important but seems to be more appropriate somewhere in the supplementary after adding the labels for the parts that are unlabeled.

7. Figure 1: Panels C-H are not even referenced individually in the text, so I would suggest either

moving them to the supplementary or simply getting rid of them.

8. Figure 1: Panel B is the most important figure of this entire work and yet is shown so tiny that the separate parts of it could be seen only under the electron microscope. I would like to suggest to the authors to enlarge this panel to the page width and make it large and clear. Also, what is really missing from this figure are the snapshots of the representative parts of the cryoEM maps showing how well different parts of this ribosome are actually resolved.

9. Correct me if I am wrong, but what is shown in Figure 2 could also be inferred from the secondary structure of rRNA, right? In other words, why do we need a structure if we can get the same conclusion just from the sequence analysis? It would be great if the authors could also explain this in the main text. Also, it would be great to show the actually observed cryoEM maps for the depicted helices so that it will become more obvious to the reader that, indeed, the base-pair interactions are lost at the bottom of the H18 in the current structure.

10. A figure that is really missing from this study is the visual comparison of this new structure of the microsporidian ribosome with (i) eukaryotic ribosome, (ii) mitochondrial ribosome, (iii) bacterial ribosome, (iv) other known structures of microsporidian ribosomes. Also, comparing their functional centers, such as the decoding center, the peptidyl transferase center, and the peptide exit tunnel, would also be super interesting to the target audience of this study. Moreover, it is curious to see how the tRNA molecules would look like in such a tiny ribosome – did the authors try to model the tRNAs in their structure? Potentially, this all could be the main-text figure 2.

11. In a few places throughout the text, the authors mention that the newly evolved ribosomal proteins substitute the lost pieces of rRNA. In regards to this, it would be interesting to check if similar substitutions also took place in mitochondrial ribosomes or not.

12. Finally, out of many topics for discussion, one that was particularly avoided is the inhibition of ribosomes by small molecules. Given that microsporidia are human parasites, it might be interesting for the general reader to see if any of the most common protein synthesis inhibitors are expected to act upon microsporidian ribosomes.

We would like to thank our reviewers for their time and feedback that helped us improve our work and prepare this improved version of the manuscript. Below, we provide a point-by-point response in which we have addressed each of their comments. We have highlighted reviewers' comments in bold and our responses in yellow.

Reviewer #1 (Remarks to the Author):

The manuscript by Nicholson et al. reports a structural investigation aimed at characterizing one of the smallest known cytosolic eukaryotic ribosomes from the parasitic microorganism *Encephalitozoon cuniculi* possessing one of the smallest and most reduced eukaryotic genomes in nature. This study follows up on two similar studies from the Klinge (Nature microbiology 2019) and the Barandun (PLOS Biology 2020) groups describing the structures of the cytosolic ribosome from two other species of microsporidians, *Paranosema locustae* and *Vairimorpha necatrix*. From an evolutionary view point, the study is certainly interesting and reports several novel observations regarding the molecular adaptation of the ribosomes from these peculiar organisms, such as the utilization of small molecules to mimic and compensate for degeneration of various protein and rRNA features. Other striking observation is the persistence of bulgeless rRNA segments and the development of "molten rRNA linkers" that link various rRNA segments through loosely pairing singles strands instead of regular rRNA helices. The authors coin the expression "lose-to-gain evolution" to characterize this revealed pattern of molecular adaptation/evolution.

The manuscript is well written and the figures are clear, in addition of being aesthetic. The references are sufficient and methods are appropriately detailed. I have no major concerns and warmly recommend the publication of this elegant study. I only have minor comments:

1. The authors should provide model vs. map curves as well, in order to appreciate the model fitting with the nicely detailed maps of the study.

To address this comment, we have added the model vs. map curves. These are now shown in Supplementary Figure 3 of the revised manuscript.

2. To fully fathom the reduction of the rRNA and to compare it with other reduced rRNAs and to yeast rRNA that is used as a reference, it is capital to provide a 2D structure diagram of the *E. cuniculi* rRNA! Ideally side-by-side with *P. locustae* or *V. necatrix* and yeast.

To address this comment, we have added a schematic comparison of 2D structure diagrams of *E. cuniculi*, *P. locustae*, *V. necatrix* and *S. cerevisiae* rRNA in the revised Figure 1.

3. Figure S2: Local resolution panels show the resolution/color scale with probably disordered numbers (2.3 – 2.9 – 2.6 Å), perhaps the correct order is 2.3 – 2.6 – 2.9 Å?

We thank the reviewer for spotting this, and have corrected the error.

4. The authors discuss the existence of various small exo-ribosomal molecules such as a possible spermidine and they illustrate it by Figure S7. Although it is rather clear, perhaps the authors would want to consider some transparency in the mesh of the lower panels and some smoothing (increased sampling/number of mesh), as it stands the mesh is quite opaque. This is a purely aesthetic comment.

We are happy to accept this excellent suggestion, and have remade Figure S7, increasing mesh transparency.

In the discussion, the authors compare the evolution of mitoribosomes to cytosolic ribosomes of microsporidians, stating that they followed similar evolutionary paths mainly regarding the rRNA reduction. I don't agree with this conclusion, as the diversity of mitoribosomes clearly showed diverse ways of evolutions. Indeed, some mitoribosomes have evolved to gain more rRNA such as yeast and more drastically, plants very widely. Indeed, plants mitoribosomes count nearly 20 to 30% more rRNA mass compared to bacteria. Other mitoribosomes are on the other extreme of the spectra and have indeed lost a significant mass of rRNA, such as mammalian mitoribosomes and the much more extreme case of kinetoplastids, especially in the small ribosomal subunit. And to make things even more complicated, some mitoribosomes have evolved to be assembled from truncated pieces of rRNA, like in many species of algae where a dozen specific mitoribosomal proteins appeared to "glue" the different pieces of rRNA. Finally, in all these mitoribosomes, the acquisition of numerous species-specific proteins is characteristic, several folds more novel proteins compared to microsporidia cytosolic ribosomes. It is believed that the evolution of mitoribosomes commenced with an RNA-constructive phase where more RNA was acquired, thus calling for the acquisition of more proteins to stabilize and accompany the extra mass of RNA, followed by a slow and progressive RNA-destructive phase where in part under the pressure of more compact mitochondrial genomes, rRNA decreases leaving the acquired proteins to ensure structural support and regulatory functions that once were performed by rRNA. I simply recommend the authors to remove this part of their discussion, or at least reduce it to the simple comparison with some mitoribosomes.

To address this comment, we have deleted the corresponding section of the discussion.

Reviewer #2 (Remarks to the Author):

Intracellular pathogens often undergo genomic reduction, but the molecular consequences of this reductive evolution are not well understood. The microsporidian *Encephalitozoon cuniculi* is a drastically reduced intracellular parasite, having one of the smallest known eukaryotic genomes, making this species an excellent system to study the effect genomic reduction on protein function. In this manuscript, the authors characterize and analyze a high-resolution cryo-EM structure of the *E. cuniculi* ribosome. Several microsporidian ribosomal structures have been examined by cryo-EM, but *E. cuniculi* is the most reduced of these and analysis of this ribosomal structure revealed numerous structural adaptations. The authors describe these novel features including loss of conserved RNA bulges, the formation of

truncated and linearized rRNA segments to minimize the length of rRNA bridges, and the binding of small molecules. Additionally, the authors identified a novel ribosomal protein and found that eL14, which was previously thought to be lost, was retained but hard to detect by sequencing homology. Together, this is a very interesting and thorough study that describes innovative features that occur in an extremely derived ribosome. Although biochemical testing of the features observed from the structure would strengthen the study, the high resolution of the structure and the comparison to other structures allows these observations to stand on their own.

Main points:

1. It is interesting that nucleotide binding was observed in *E. cuniculi* and *P. locustae*, but not in *V. necatrix*. Why is that? Are the F170 and K172 residues not present in *V. necatrix* or is there some other explanation?

We agree. This nucleotide absence in *V. necatrix* ribosomes is puzzling, especially given the presence of the F170 and F172 residues in *V. necatrix* eL20. To address this comment and provide a possible explanation for this absence, we compared cryo-EM maps and models of *E. cuniculi*, *P. locustae* and *V. necatrix* ribosomes and added the following paragraph to the discussion section:

“It is unclear why the nucleotide binding was observed in *E. cuniculi* and *P. locustae* ribosomes but not in *V. necatrix* ribosomes, despite the presence of residues F170 and K172 in *V. necatrix* protein eL20. This absence may be caused by residue 43 of uL6 (located near the nucleotide-binding pocket), which is a tyrosine in *V. necatrix*, rather than threonine, as in *E. cuniculi* and *P. locustae*. It is possible that the bulky aromatic sidechain of Tyr43 prevents nucleotide binding due to steric overlap. Alternatively, the apparent nucleotide absence could be caused by the lower resolution of the cryo-EM map, which precluded accurate modeling of that segment of *V. necatrix* ribosomes.”

E. cuniculi

P. locustae

V. necatrix

2. For Figure 4D, it would be useful to add the sequences of *P. locustae* and *V. necatrix*. It would also be useful to add a Nematocida species. Labeling amphiamibly and mitosporidium as they are in C would be more informative than labeling them “other eukaryotes”. The figure legend states that “illustrate that the nucleotide-binding residues F170 and K172 are conserved across all microsporidian species,

except for the early-branched microsporidia”, but this appears to not be the case as *S. lophii* does not contain these residues.

To address this comment, we have corrected this panel, adding the missing sequences of eL20 homologs and changing the labels of the eukaryotic species. To make the figure legend more accurate, we now state that “residues F170 and K172 are conserved in most canonical microsporidian species, aside from *S. lophii*”.

3. On line 215, it is mentioned that the Asp51 and Arg56 mutations in uL15 are microsporidia-specific. However, in figure S7, these mutations are only shown for *E. cuniculi*. It is unclear whether these mutations are conserved amongst microsporidia, or only occur in *E. cuniculi*.

Indeed, we did not support this statement in the initial version of our manuscript. To address this comment, we have added Supplementary Data S3, showing aligned sequences of eukaryotic uL15 homologs. We then added a new panel to the Supplementary figure S7 (Supplementary Figure S8 in the revised manuscript) to illustrate microsporidia-specific occurrence of the Asp51 and Arg56 residues.

Minor point:

1. The dormancy factor MDF1 is mentioned in figure 1 and figure S5, but not in the main text. It would be good to add to the main text description of the results related to MDF1.

To address this comment, we moved the Supplementary Figure S5 to the main text, describing ribosome hibernation by Mdf1.

Reviewer #3 (Remarks to the Author):

In the presented work by Nicholson et al., the authors reported a new cryo-EM structure of the functionally inactive dormant ribosome from one of the smallest eukaryotes *Encephalitozoon cuniculi*, which belongs to the microsporidia group. This is not the first structure of the microsporidian ribosome and definitely not the first structure of the eukaryotic ribosome. However, in comparison to the previously reported two structures of ribosomes from microsporidia, this one appears to have the highest resolution reported to date. Moreover, given that the ribosomes from this species of microsporidia underwent one of the largest known reductions of size, this structure is impactful from a fundamental point of view because it can shed light on the evolution of ribosomes in this group of intracellular parasites of eukaryotes.

Overall, this work is presented as a very concise and clearly written manuscript with beautiful and stylish illustrations that definitely capture an eye. The strong side of this work is the better structural model of the microsporidian ribosome, which in addition are probably very hard to isolate and purify. However, there are several major concerns denoted below that need to be addressed by the authors.

Major issues:

1. Authors claim that they identified AMP and spermidine-like molecules stably bound to the microsporidian ribosome. I do agree that the cryoEM map that is shown in Figure 4A does resemble AMP in shape, but the structural methods (unless we are talking about atomic resolution structures at 1-1.4Å) are not the methods typically used to identify the molecules. The authors should provide mass-spec data or some other biochemical data to add more credibility to their assignments of AMP.

We agree with the reviewer that our description of the small molecules was somewhat misleading, giving an impression that not only we could identify the small molecules, but also reveal their identity. We did not want, however, to confuse our reader, especially because the identity of the small molecules is not relevant for our major finding.

Our major finding is that microsporidian parasites use small molecules as ribosomal building blocks to compensate for the absence of large chunks of rRNA and ribosomal proteins. However, as the reviewer notes, ribosome isolation from microsporidia is highly challenging, precluding many experiments that can be routinely done using ribosomes from free-living species. In contrast to free-living species, *E. cuniculi* cannot grow *in vitro* and instead grow only inside host cells. Their growth is slow, and their amounts are scarce even with large quantities of biological material, multiple weeks of cultivation, and laborious maintenance to ensure that the parasitic infection does not kill the host cells in the infected culture. We in fact attempted mass-spec and radiolabel-based identification of the small molecules before submitting our work to bioRxiv, but could not obtain conclusive data due to the insufficient quantities or purity of *E. cuniculi* ribosome samples that we can realistically produce today. Thus, whilst determining the identity of these molecules is highly desirable, and is a key goal in our future research, we feel it is a prodigious undertaking that is beyond the scope of *this* study.

We, therefore, focused our work on findings in which the identity of these small molecules is not necessary for conceptually novel insights. Specifically, to address this comment and make sure that we do not mislead our reader, we changed the “AMP-like molecule” to the “nucleotide-like molecule” throughout the manuscript and figures. We then deleted all the speculations about the impact of the nucleotide binding on microsporidia biology.

2. Regardless of whether that extra density corresponds to AMP or not, typically, it is OK to speculate and make assumptions. However, building new assumptions on assumptions is not a good practice. In other words, what if that density is not an AMP? Why couldn't it be a GMP molecule, for example? What if this is something else? Then all the discussions about its new role in sustaining the structure of the microsporidian ribosome do not have sense anymore. It is very dangerous to make such second-level assumptions without providing compelling experimental evidence.

We thank the reviewer for drawing our attention to these second-level assumptions in our manuscript. To address this comment, we deleted these assumptions about the possible implications of nucleotide binding for microsporidia biology.

3. Although the manuscript indeed is nicely written, the style appeared somewhat more like a review rather than an experimental paper. I think the authors should describe in more detail the peculiarities of ribosome purification and structure determination rather than speculating about possible evolution on every page. This work is abundant with comparisons of various small structural features of the ribosome, which is important (no doubt) but is missing comparisons of the global structures of the ribosomes from different groups of organisms with the current one.

To address this comment, we added an opening paragraph in our results section, describing ribosome purification. We also changed Figure 1 to compare *E. cuniculi* ribosomes with other eukaryotic ribosomes, focusing on the conservation of the active centers and showing the cryoEM density to illustrate the accuracy of our 3D model.

Minor issues:

4. At the very beginning of the results section, the authors should elaborate more on how ribosomes were purified from the spores. I guess it is not really a trivial task to purify ribosomes from microsporidian spores, and, therefore, it is likely to be interesting to the reader. Moreover, this is definitely not something that is routinely done in the labs, and therefore it is important to emphasize how difficult it could be. Yes, of course, the details of it are in the methods, but the story will likely benefit from adding a few sentences explaining how purification was done and what are the typical pitfalls of this procedure. I would also recommend adding a few sentences here explaining how the structure was determined, how particles were classified, what masks were applied to which parts of the ribosome, etc.

To address this comment, we have now expanded the results section by opening our results with a section that describes our ribosome purification and structure determination section procedures.

5. Because the ribosomes were isolated from the spores, apparently, they represent a dormant, inactive state, which is mentioned in a couple of places. I think this is a very important aspect that this structure does not represent a functionally active state of the ribosome from microsporidia and should be explicitly stated in the main text.

To address this comment, we added the results section "Hibernation of *E. cuniculi* ribosomes with the protein Mdf1". We also moved our supplementary Figure 5 to the main text to explicitly state that we observe hibernating ribosomes in complex with the hibernation factor Mdf1.

6. Figure 1: Panel A is important but seems to be more appropriate somewhere in the supplementary after adding the labels for the parts that are unlabeled.

To address this comment, we modified Figure 1 to incorporate this scheme into one of the figure panels.

7. Figure 1: Panels C-H are not even referenced individually in the text, so I would suggest either moving them to the supplementary or simply getting rid of them.

To address this comment, we moved these panels to Figure 8, and we reference each panel individually in the discussion section.

8. Figure 1: Panel B is the most important figure of this entire work and yet is shown so tiny that the separate parts of it could be seen only under the electron microscope. I would like to suggest to the authors to enlarge this panel to the page width and make it large and clear. Also, what is really missing from this figure are the snapshots of the representative parts of the cryoEM maps showing how well different parts of this ribosome are actually resolved.

To address this comment, we modified this panel, supplementing it with snapshots of cryo-EM maps.

9. Correct me if I am wrong, but what is shown in Figure 2 could also be inferred from the secondary structure of rRNA, right? In other words, why do we need a structure if we can get the same conclusion just from the sequence analysis? It would be great if the authors could also explain this in the main text. Also, it would be great to show the actually observed cryoEM maps for the depicted helices so that it will become more obvious to the reader that, indeed, the base-pair interactions are lost at the bottom of the H18 in the current structure.

This is not entirely correct. While sequence analyses can indeed provide some valuable assumptions about the tertiary structure of RNA molecules, the actual structural studies—including ribosomes from different species—continue to defy many of these sequence-based assumptions. It turns out that most rRNA segments, which appear to be single-stranded in the secondary structure diagrams, often form highly ordered 3D structures. These structures involve unexpected base-pairing between distant rRNA segments and the formation of kissing loops, non-canonical pairs, or pairs between rRNA bulges, etc. For example, in eukaryotic ribosomes, the rRNA expansion segments ES3S and ES6S were predicted to be single-stranded. However, the actual structural studies revealed a kissing loop between ES3S and ES6S, showing that these seeming single-stranded segments form a nearly-ideal A-form helix. This is just one of many examples where predictions do not accurately capture the 3D structure of RNA.

10. A figure that is really missing from this study is the visual comparison of this new structure of the microsporidian ribosome with (i) eukaryotic ribosome, (ii) mitochondrial ribosome, (iii) bacterial ribosome, (iv) other known structures of microsporidian ribosomes. Also, comparing their functional centers, such as the decoding center, the peptidyl transferase center, and the peptide exit tunnel, would also be super interesting to the target audience of this study. Moreover, it is curious to see how the tRNA molecules would look like is such a tiny ribosome – did the authors

try to model the tRNAs in their structure? Potentially, this all could be the main-text figure 2.

To address this comment, we added a new panel to Figure 1. This panel compares *E. cuniculi* ribosomes with ribosomes from other microsporidian species and yeasts. We then added Figure 8, which provides a schematic outline of ribosome evolution, comparing microsporidian ribosomes with ribosomes from bacteria and other eukaryotes, such as yeasts and humans. We also added additional panels of Figure 1 to show high conservation of the active centers of *E. cuniculi* ribosome compared to other eukaryotic ribosomes. We do agree with the reviewer that it was important to make such a statement in the text.

11. In a few places throughout the text, the authors mention that the newly evolved ribosomal proteins substitute the lost pieces of rRNA. In regards to this, it would be interesting to check if similar substitutions also took place in mitochondrial ribosomes or not.

We have removed all the description of the mitochondrial evolution, following the major recommendation of the reviewer #1.

12. Finally, out of many topics for discussion, one that was particularly avoided is the inhibition of ribosomes by small molecules. Given that microsporidia are human parasites, it might be interesting for the general reader to see if any of the most common protein synthesis inhibitors are expected to act upon microsporidian ribosomes.

To address this comment, we compared the decoding center and the peptidyl-transferase center of *E. cuniculi* ribosomes with these centers in yeast ribosomes and showed these comparisons in Figure 1. We have also added an additional section in the results of our manuscript to describe microsporidia-specific variations in the decoding center/aminoglycosides binding site.

Reviewers' Comments:

Reviewer #1:

Remarks to the Author:

The authors have addressed appropriately all of my points. I have no more comment to raise.

Reviewer #2:

Remarks to the Author:

The authors have adequately addressed all of my concerns and I now enthusiastically support the publication of the article.

A minor comment is that in response to reviewer three the authors say they now have included a figure 8 that has comparison of microsporidia, bacteria, yeast, and human ribosomes. This figure does not appear to be currently present in the manuscript.

Reviewer #3:

Remarks to the Author:

After carefully reading the revised version of the manuscript as well as the responses of the authors to the reviewer's comments, I need to say that the authors accomplished an excellent job of improving the manuscript and addressing the critical point raised by this reviewer. I think that most of my critical points are satisfactorily addressed and now this work appears as a coherent and impactful story that definitely deserves publication.

The only minor comment I still have is to remove the word "atomic" from line 120 and throughout the text. Although 2.7Å is an awesome resolution for a cryo-EM ribosome structure, individual atoms cannot be seen at this resolution and, therefore, it cannot be called atomic.

This file contains our separate point-by-point response to the reviewers' comments, reproduced verbatim.

Reviewer #1 (Remarks to the Author):

The authors have addressed appropriately all of my points. I have no more comment to raise.

We thank the reviewer for this appreciation of our efforts to improve this work.

Reviewer #2 (Remarks to the Author):

The authors have adequately addressed all of my concerns and I now enthusiastically support the publication of the article.

A minor comment is that in response to reviewer three the authors say they now have included a figure 8 that has comparison of microsporidia, bacteria, yeast, and human ribosomes. This figure does not appear to be currently present in the manuscript.

We thank the reviewer for spotting this typo in our response. To avoid misunderstanding, we have included these comparisons in the updated Figure 1 (instead of Figure 8) to show comparison of rRNA reduction in microsporidian species, and the comparison of the decoding center ribosomes from bacteria, canonical eukaryotes (such as yeasts and humans), and microsporidia.

Reviewer #3 (Remarks to the Author):

After carefully reading the revised version of the manuscript as well as the responses of the authors to the reviewer's comments, I need to say that the authors accomplished an excellent job of improving the manuscript and addressing the critical point raised by this reviewer. I think that most of my critical points are satisfactorily addressed and now this work appears as a coherent and impactful story that definitely deserves publication.

The only minor comment I still have is to remove the word "atomic" from line 120 and throughout the text. Although 2.7Å is an awesome resolution for a cryo-EM ribosome structure, individual atoms cannot be seen at this resolution and, therefore, it cannot be called atomic.

We have now removed the word atomic from the manuscript's text.